# Human Papillomavirus and Cellular Pathways: Hits and Targets

**DOI:** 10.3390/pathogens10030262

**Published:** 2021-02-25

**Authors:** Alessandro Medda, Daria Duca, Susanna Chiocca

**Affiliations:** Department of Experimental Oncology, IEO, European Institute of Oncology IRCCS, 20139 Milan, Italy; alessandro.medda@ieo.it (A.M.); daria.duca@ieo.it (D.D.)

**Keywords:** HPV, EGFR, PI3K/Akt/mTOR, AP-1, autophagy, EMT, JAK/STAT, DNA damage response, miRNA, head and neck cancer, cervical cancer

## Abstract

The Human Papillomavirus (HPV) is the causative agent of different kinds of tumors, including cervical cancers, non-melanoma skin cancers, anogenital cancers, and head and neck cancers. Despite the vaccination campaigns implemented over the last decades, we are far from eradicating HPV-driven malignancies. Moreover, the lack of targeted therapies to tackle HPV-related tumors exacerbates this problem. Biomarkers for early detection of the pathology and more tailored therapeutic approaches are needed, and a complete understanding of HPV-driven tumorigenesis is essential to reach this goal. In this review, we overview the molecular pathways implicated in HPV infection and carcinogenesis, emphasizing the potential targets for new therapeutic strategies as well as new biomarkers.

## 1. Introduction

The Human Papillomavirus (HPV) is the causative agent of more than 90% of cervical cancers, but it is also implicated in the development of other malignancies, such as non-melanoma skin cancers, anogenital tumors, and squamous cell carcinoma of the Head and Neck (HNSCC) [1]. Despite the presence of vaccines against the most prevalent HPVs, the burden of HPV-related tumors is still far from being in consistent reduction [2]. Moreover, the lack of targeted therapy against HPV-related cancers evidences the need of new therapeutic approaches. Thus, a complete understanding of the pathways involved in HPV-mediated carcinogenesis could open to new strategies for targeting these tumors.

In this review, we will give a broad overview of the molecular mechanisms and pathways affected by HPV as well as which of them has potential features for new biomarkers or targeted therapy.

HPVs are small DNA viruses belonging to the *Papillomaviridae* family [3]. Currently, more than 200 HPV types have been defined and can be subdivided in two main groups: the high risk (HR) and low risk (LR), based on their ability to induce cancers. Indeed, LR HPVs are responsible for anogenital or cutaneous warts, recurrent respiratory papillomatosis and Heck’s disease, while HR HPV are known for their ability to drive tumorigenesis in the cervix, in the anogenital tract and in the mucosa of the Head and Neck (HR and LR HPVs and their related diseases are listed in Table 1) (reviewed in [4]). HPVs are non-enveloped viruses consisting of an icosahedral capsid of about 60 nm in diameter, with a double stranded circular DNA of approximately 8000 base pairs [5]. HPVs contain three genomic regions, including approximately ten open reading frames (ORFs). Polycistronic mRNAs generate many of the viral proteins [6]. The viral genome can be subdivided into three regions, including the early region (E), with up to seven ORFs encoding viral regulatory proteins; the late region (L), that encodes the two viral capsid proteins; and the long control region (LCR), or upstream regulatory region (URR), composed by the origin of replication and transcription control sequences [7].

HPV infects basal cells of the mucosal epithelium. Capsid protein L1, upon attachment to heparan sulphate proteoglycans (HSPG), changes conformation and exposes capsid protein L2 to cleavage, inducing internalization of the virus. E1 and E2 proteins are the first transcribed and are essential for viral genome amplification, which can stay in the basal cells either as a multicopy plasmid or episome, or a mix of both. E1 and E2 recruit cellular components of DNA replication machinery to the replication fork and bind to the origin of replication. The E5 protein is a multi-pass protein which activates receptors tyrosine kinase (RTKs) and induces proliferation. The E6 and E7 proteins change the environment of differentiated basal cells of the epithelium. Specifically, E6 inhibits apoptosis by interacting with p53 and inducing its degradation. E6 increases telomerase activity by upregulation of telomerase reverse transcriptase (TERT). E7 protein interacts with the retinoblastoma protein (pRb), sending it to proteasomal degradation, thereby releasing and activating important transcription factors involved in cell cycle progression. E6 and E7 modify the cellular environment to induce genome amplification in growth arrested differentiated cells, and, through induction of uncontrolled proliferation, they increase the infected area. HPV genome can then be packaged into L1 and L2 capsid protein and, after a maturation period, can exit from cells which have lost nuclear and cytoplasmic integrity, aided by the E4 protein that disrupts cytokeratin filaments (reviewed in [4]).

The main drivers of carcinogenesis, which discriminates HR and LR HPVs, are the early proteins E6, E7, and E5. Both HR and LR E6 and E7 proteins can interact respectively with p53 and retinoblastoma protein (pRb), but only HR HPVs are able to induce their degradation and inactivation. Upon HPV integration into the host genome, E6 and E7 lose their regulation mediated by the E2 repressor protein, leading to their uncontrolled expression. E6 and E7 oncoproteins expression induces genomic instability and accelerates the accumulation of mutations, hence resulting in the development of malignancies [8,9].

To precisely assess whether a tumor is truly HPV-driven, several markers are used for HPV detection: viral DNA detection through PCR techniques, E6/E7 HPV mRNA RT-PCR, HPV DNA in situ hybridization, and p16^INK4a^ detection through immunohistochemistry [10]. This is crucial, for example, in HPV-related head and neck cancers, where HPV positive tumors have a different prognosis with respect to HPV negative ones (recently reviewed in [11]).

In the next sections, we will give an overview of the main pathways and cellular processes affected by the action of HPV, indicating potential biomarkers for early detection of HPV-driven tumors, as well as potential druggable targets for antiviral and antineoplastic agents.

## 2. Signaling Pathways and HPV

### 2.1. p53

The ternary E6/E6AP/p53 is the best characterized interaction between HPV and host proteins [12]. The tumor suppressor p53 is essential in preventing tumors thanks to its diverse functions. P53 is a transcription factor involved in many cellular processes, including induction of DNA damage response, cell cycle arrest, and apoptosis [13]. To evade the control of the genome achieved by p53, it is mutated in 50% of human cancers [14].

HR-HPVs are able to induce p53 degradation, as well as escape apoptosis and cell cycle arrest. The viral oncoprotein E6 forms a ternary complex with p53 and with the ubiquitin E3 ligase E6-associated protein (E6AP), resulting in the ubiquitination of p53 and its consequent degradation by the proteasome system (Figure 1a). It is important to note that only high risk, and not low risk HPVs, can cause p53 degradation [15,16]. p53 levels are kept low and infected cells can evade apoptosis and cell cycle arrest. This also causes chromosomal instability, which eventually leads to carcinogenesis [17,18]. In this context, it is easy to understand the importance of p53 downregulation for carcinogenesis and why the disruption of the binding between E6 and p53 could be a potential target for HPV-mediated cancer therapy. Indeed, recently, Celegato et al. identified a small molecule inhibitor of p53/E6 interaction, which repristinates p53 activity and blocks cancer cells growth, thus showing a promising target for HPV-related cancers [19]. HR HPV E6/E7 oncoproteins can also upregulate, at both the mRNA and protein level, another member of the p53 family of proteins, namely p63 and in particular the ΔNp63α isoform, which is important in HNSCC carcinogenesis [20]. This could allow the targeting of p63 for the treatment of HPV-positive HNSCC patients, for example, with the use of histone deacetylase inhibitors which were shown to induce its downregulation in HNSCC cell lines [21].

### 2.2. pRb and Pocket Proteins

The tumor suppressor retinoblastoma protein is a member of the so-called “pocket proteins”, together with p107 and p130 [22,23,24,25]. Pocket proteins are fundamental in controlling the cell cycle by directly interacting with E2F family proteins (reviewed in [26]). In a hypophosphorylated form, pRb binds to E2F, causing its negative regulation and a quiescent state of the cell. When mitogenic signals are present, D-cyclins are transcribed and associate with cyclin dependent kinases (CDKs), specifically CDK4 and CDK6, which phosphorylate pRb. Hyperphosphorylated pRb releases to E2F that, now free and active, can transcribe a family of genes involved in cell cycle progression [27].

The HPV oncoprotein E7 interacts with pRb, inducing the release of E2F and leading to uncontrolled cell cycle progression. HR-HPV E7 is able to inactivate and destabilize pRb even in the absence of CDKs and to induce its degradation through the proteasome system (Figure 1b) [28,29,30]. LR HPV 6 and 11 express E7 proteins with a lower pRb binding efficiency and without transforming activity in vitro [31,32]. On the other hand, the LR HPV1 E7 has high affinity to pRb as HR HPVs, but it fails to induce transformation of primary cells [32]; moreover, it is not capable of inducing pRb degradation [1].

### 2.3. EGFR

The epidermal growth factor receptor (EGFR) is a tyrosine kinase, member of the ErbB/HER (ERBB, from the related avian viral erythroblastosis oncogene; HER, human EGF receptor) family [33]. EGFR is a transmembrane protein that is activated by the binding of some ligands, in particular the epithelial growth factor (EGF), the transforming growth factor α (TGF α), and others [34]. It consists of an extracellular part that binds the ligands, a transmembrane part, and an intracellular part capable of catalytic activity. Inactive EGFR is monomeric, but the binding of ligands activates it, giving rise to the formation of homodimers. EGFR is a very well-studied proto-oncogene, because it is implicated in many cellular processes such as proliferation, migration, survival, and angiogenesis [35,36,37]. Once activated, EGFR homodimers autophosphorylate, and propagate extracellular mitogenic signals to the nucleus, resulting in the activation of many cellular genes and pathways such as mitogen-activated protein kinase (MAPK) and phosphoinositide-3-kinase (PI3K)/protein kinase B (AKT) implicated in differentiation, mitogenesis, mobility, and survival [35]. EGFR is important in cancer cell proliferation: it regulates many metabolic processes (fatty acids and pyrimidines synthesis, glucose catabolism) in a direct fashion by phosphorylating enzymes, or indirectly by activating signaling pathways (AKT) [38,39,40].

High EGFR expression is associated with poor prognosis in cervical cancer [41]. The HPV oncoprotein E5 is involved in the activation and increase of the EGFR pathway dependently on the ligand (Figure 1c) [42]. Viral oncoprotein E5 can upregulate VEGF (vascular endothelial growth factor) and cyclooxygenase 2 through EGFR [43]. Activation of EGFR pathway by E5 results in the initiation of an intracellular cascade that activates many proto-oncogenes. In particular, mitogen associated protein kinases (MAPKs) and the activating protein-1 (AP-1) components are activated, inducing the expression of the viral oncoproteins E6/E7, as explained in detail in the following sections [42,44]. Moreover, HPV16 E5 enhances recycling of the EGFR to the surface of cells and an increase in EGFR phosphorylation levels, although requiring EGF binding [45,46]. This can be observed in the absence of any change in EGF internalization and degradation rates, as well as affinity levels of EGFR for EGF, resulting in an increased number of EGF receptors on the cell surface and lower degradation of EGF-bound EGFR. During HPV infection, E5 has a key role in hyperproliferation of keratinocytes, enhancing EGFR signaling to delay differentiation [42]. E5 is expressed mainly in differentiating suprabasal cells of the stratified epithelium, based on in vivo and in vitro studies [47]. EGFR is an important target for targeted therapy in HNSCC. High expression of the receptor, detected by immunohistochemistry (IHC), occurs in 90% of HNSCC specimens. Many studies have correlated high EGFR expression with low survival rates, radioresistance, and locoregional failure. Inhibition of EGFR confers higher sensitivity of cancer cells to ionizing radiations in preclinical studies on HNSCC [48,49,50,51]. Cetuximab, the monoclonal antibody against EGFR, is currently used for recurrent and metastatic HNSCC [52]. However, it is not clear if there is a difference in efficacy between HPV positive and HPV negative tumors because results of different studies are controversial [53,54].

### 2.4. PI3K/Akt/mTOR

The PI3K/Akt/mTOR cascade is important for cellular control and signal transduction: it promotes cell survival, growth, proliferation, migration, and energy metabolism. Phosphoinositide 3-kinases (PI3Ks) are a class of enzymes that are phosphorylated (activated) in the presence of external stimuli and are regulated by receptor tyrosine kinases (RTKs) or G protein-coupled receptors (GPCRs) GTPases. Active PI3K generates the membrane lipid phosphatidylinositol-3,4,5-trisphosphate (PIP3) and phosphatidylinositol-3,4-bisphosphate (PI3,4P2) by phosphorylation of phosphatidylinositol-4,5-bisphosphate (PI4,5P2) [55]. This induces the recruitment of Akt to the cell membrane and the consequent activation of this protein through its phosphorylation by the mammalian target of rapamycin complex 2 (mTORC2). Phosphatase and tensin homolog (PTEN) negatively regulates PI3K activation of Akt by dephosphorylation of PIP3. Active Akt phosphorylates many targets involved in cell cycle control, cell proliferation, cell mobilization, angiogenesis, anti-apoptosis, and cell survival [56]. Among those targets, Akt inhibits tuberous sclerosis complex 2 (TSC2), with the consequent activation of mTORC1. mTORC1 is involved in sensing of energy, oxygen, growth factor, amino acids and stress; this to ensure adequate resources to activate downstream processes. Phosphorylated mTORC1 activates protein translation, lipid and nucleotide synthesis, and inhibits autophagy (reviewed in [57]).

The PI3K/Akt/mTOR axis is frequently deregulated in many tumor types, contributing to malignant growth and resistance to therapy [58]. PIK3CA gene (encoding class I PI3K catalytic subunit) is mutated or amplified in many cancers; in particular, it is mutated in 17.5% and amplified in 15.7% of HNSCC; the loss of PTEN also contributes to carcinogenesis [59]. Moreover, mutations in PIK3CA and PTEN are more common in HPV-positive than HPV-negative HNSCC [59,60]. The HPV16 E7 protein binds to protein phosphatase 2A (PP2A) subunits, preventing their interaction with p-Akt and keeping it active. E6 can activate Akt as well, or bind TSC2, leading to its degradation and resulting in stimulation of mTORC1 (Figure 1d) [61,62,63,64]. Furthermore, targeting mTOR inhibits carcinogenesis in a mouse model of HPV [65]. Recently, many clinical trials on HNSCC patients are ongoing to evaluate the efficiency of PI3K/AKT/mTOR inhibitors, often using rapamycin analogs, and these show partially promising treatment responses [66,67,68]. Thus, this pathway can be considered a potential target for the treatment of HPV-induced cancers.

### 2.5. JNK/ERK/ AP-1

#### 2.5.1. JNK

c-jun N-terminal kinase (JNK) is a subfamily of Ser/Thr kinases from the canonical signal transduction of MAPK; JNK1, JNK2, and JNK3 are present, encoded by three different genes [69,70]. JNKs respond to different external signals, such as infections (both viral and bacterial), cytokines, growth factors, heat shock, UV radiation, and other stresses [71]. JNKs are activated by a cascade of upstream signals (JNK kinases and JNK kinase kinases) and, in turn, phosphorylate target proteins, including activating transcription factors (ATF, ETS Like-1 protein (Elk1) and AP-1 family proteins [72,73,74]. Among AP-1 family members, JNK phosphorylates Jun protein family. It has been shown that JNK1/2 phosphorylation is upregulated in primary keratinocytes transduced with HPV18 [75]. Moreover, a recent study found that HPV E6 induces JNK phosphorylation via the PDZ-binding motif, activating c-jun expression, thereby promoting proliferation and expression of viral oncoproteins through EGFR in cervical cancer [76].

#### 2.5.2. ERK

The extracellular signal-regulated kinase (ERK) pathway is implicated in the phosphorylation of a variety of substrates involved in cell proliferation, differentiation, survival, and motility [77]. The Ras (from Rat sarcoma) GTPase recruits RAF (Rapidly Accelerated Fibrosarcoma), in particular Raf-1, which in turn phosphorylates serine of MEK1/2 (MAPK/ERK kinase 1 and 2) [78,79,80]. MEK1/2 induce ERK1/2 phosphorylation of tyrosine and threonine residues, resulting in the activation of a plethora of downstream pathways [77]. For the variety of pathways regulated by ERK signaling, the deregulation of this pathway has been associated with different kinds of cancers [81,82]. Nowadays, many inhibitors of the ERK signaling pathway are available, increasing the interest in this pathway for targeted cancer therapy [83].

The ERK pathway is involved in HPV-induced cancers. The 5-aminolevulinic acid photodynamic therapy downregulates HPV viral load by the ERK, Akt, and mTOR pathways [84]. A recent report has shown that the E6 protein can change the activity of the Eukaryotic translation initiation factor 4E (eIF4E) protein via ERK and Akt pathways [85]. Moreover, the E6 oncoprotein induces the activation of ERK signaling and upregulates the expression of HIF-1α (hypoxia inducible factor 1α), VEGF, and interleukin 8 expression [86]. Activation of Erk1/2 signaling by benzo[α]pyrene upregulates the expression of HPV 31 [87].

#### 2.5.3. AP-1

AP-1 is a dimeric transcription factor implicated in the regulation of many pathways, including differentiation, proliferation, and apoptosis (reviewed in [88]). AP-1 can vary its transcriptional function according to dimer composition, which can range over 4 different family proteins: Jun, Fos, ATF/CREB (activating transcription factor, cyclic AMP-responsive element-binding), and Maf (musculoaponeurotic fibrosarcoma) [88]. AP-1 is regulated at multiple levels and in a complex way from dimer composition to specific interactions between AP-1 proteins and other transcription factors as well as to transcriptional and post-transcriptional mechanisms [89]. The most relevant proteins of the AP-1 complex are from the Jun and Fos families, which form heterodimers with the highest affinity to an asymmetric heptameric sequence TGA(C/G)TCA (called AP-1 sequence) and, with a slightly lower affinity, to a symmetric octameric sequence, TGACGTCA [90,91]. Jun family proteins can form homodimers and are composed by c-jun, junB, and junD [92]. C-jun activity is regulated by the JNKs on serine 63 and 73 [93]. The Fos family of proteins can only form heterodimers with Jun and is composed by c-Fos, FosB, Fra-1, and Fra-2. In particular, c-fos is regulated by the ERK signaling pathway with a dual mechanism: increasing c-fos transcription and increasing its activity by direct phosphorylation [94,95,96,97].

AP-1 is fundamental for HPV oncoproteins transcription. HPV 18 promoter 105 (p97 in HPV16) contains an AP-1 consensus sequence, that starts the transcription of E6/E7 by AP-1 transcription factors (Figure 1e) [98]. Mutations in the AP-1 binding site abolish E6/E7 expression, and altered AP-1 is correlated with tumorigenic phenotypes in HeLa cells, while c-fos upregulation induces cervical cancer cells proliferation [99,100]. AP-1 is a key regulator of E6/E7 expression and mediates chemoradiation resistance, which can be reverted by curcumin [101]. It has been shown that exposure to tobacco upregulates the expression of E6/E7 oncoproteins by increasing AP-1 mediated transcription in cervical cancer cells [102]. On the contrary, inhibiting the AP-1 pathway using berberine, induces the suppression of E6/E7 and the restoration of p53 and pRb activity, resulting in growth arrest and apoptosis in cervical cancer cells [103,104]. Given the importance of this pathway for HPV oncoproteins’ expression, this could be a potential target for HPV-induced cancers.

### 2.6. Autophagy

Autophagy is a self-consumption mechanism used by cells to maintain homeostasis. It balances sources of energy in response to nutrient stress. It is involved in degradation of long-lived proteins, misfolded proteins, and damaged mitochondria, as well as the elimination of intracellular pathogens [105]. It has been shown that autophagy can be deregulated in several types of cancers [106]. Autophagy can be subdivided in microautophagy, chaperon-mediated autophagy and macroautophagy (hereafter referred as autophagy). During autophagy autophagosomes, double-membraned organelles containing cargoes from different origins, upon fusion with lysosomes (acidic degradative organelles), give rise to autolysosomes, achieving degradation of the cargo [107]. Initiation of the phagophore is mediated by the unc-51 like autophagy activating kinase 1 and 2 (ULK1/2) kinase complex [108]. Autophagy factors are recruited to the phagophore and form a curved double-membrane layer that detaches from the membrane it originates from. The elongation process consists in the expansion of the phagophore by the class III PI3K complex I composed by the vacuolar protein sorting 34 (VPS34), PI3K, autophagy related gene 14L (ATG14L), VPS15, and Beclin1. Thus, two ubiquitin-like conjugation systems are recruited to conjugate phosphatidylethanolamine, to the microtubule-associated protein 1 light chain 3 (LC3) [109]. The lipidated form of LC3 (LC3-II), localized on the autophagosome, is widely used as an indicator of autophagic flux, and it regulates membrane elongation and autophagosome maturation [110]. P62, another marker of autophagy, interacts with LC3 and localizes in the autophagosomes. Fusion of autophagosomes with lysosomes is achieved by class III PI3K complex II, composed of VPS34, VPS15, Beclin 1, and UV radiation resistance-associated gene protein (UVRAG), which activates Ras-associated protein-7 (Rab7), leading to activity formation of autolysosomes [111].

HPV16 E5 down-regulates the mRNA expression of autophagic genes, such as ATG4a, ATG5, LC3, ULK1, ULK2, Beclin 1, and ATG7, suggesting a downregulation of phagophore assembly (reviewed in [112]). With a different approach, HPV16 E6/E7 affect autophagy by inhibiting autophagosome-lysosome fusion (Figure 1f). Oncoproteins’ overexpression in primary human keratinocytes upregulated both the lipidated LC3 and p62, indicating autophagosome accumulation (increase in LC3-II) caused by decreased degradation capability (increased p62) [113]. Moreover, HPV oncoprotein E7 induces the degradation of Ambra1, inhibiting autophagy and sensitizing HNSCC cells to cisplatin-induced apoptosis [114].

## 3. EMT

Epithelial-mesenchymal transition (EMT) is a phenotypical change occurring in the epithelial cells of many malignant tumors, and it is a shift in polarity corresponding to greater invasiveness and metastatic potential [115,116,117]. EMT is a normal physiologic process in wound healing and fibrosis (Type II) and embryonic development (Type I), but in tumorigenesis, it acquires a pathologic function, leading to fibrosis and cancer (Type III), which associates with a poor prognosis [118,119,120]. As the shift also favors a range of other underlying mechanisms—such as cell migration, prevention of apoptosis and senescence, and creation of an immunosuppressive microenvironment—resistance to common and advanced treatments such as chemotherapy and immunotherapy might develop [117].

At the molecular level, many changes are associated with and lead to EMT in cancer cells. Their increased invasive potential is supposedly due to HPV-16 E6 and E7 oncoproteins’ activation of Slug, Twist, and ZEB1/2 transcription factors. The action of E7 leads to actin reorganization and mutation of important cell adhesion molecules. For example, mesenchymal-related Vimentin (VIM), N-Cadherin, and Fibronectin (Fn) overcome epithelial cell-cell adhesion complexes such as E-cadherin, occludin, claudin, and β-catenin (Figure 2a) [116,121,122,123]. Several cell-signaling molecules are also essential for transformed cancer cells to evade apoptosis. Upregulation of the epidermal growth factor receptor (EGFR) by E5 and by transforming growth factor-β (TGF-β) allows escape from Fas/FasL (ligand) and Bax/Bak mediated programmed cell death [116]. The detachment from the basal membrane through proteolytic degradation, leads to the loss of the typical apico-basal orientation and a change associated with angiogenesis and metastasis. A close correlation between EMT and cancer stem cells (CSC), responsible for the heterogenicity and self-renewal of cancer cell populations, has also been observed [117].

### 3.1. E-Cadherin

Among the various changes induced by HPV, the cadherin switch is considered an important indicator of cell transformation, which, as mentioned, is associated with invasiveness and metastatic potential [116,117,124]. Cadherins are part of a superfamily of calcium (Ca^2+^) dependent membrane proteins further divided into cadherin, cadherin-related proteins, and protocadherin families whose role is mainly related to cell adhesion and various developmental differentiation processes [125,126]. Cadherin switch means a change in the normal epithelial cell adhesion molecule E-cadherin to the mesenchymal-associated N-cadherin and P-cadherin (Hu et al., 2015). These are type I classical cadherins with extracellular domains for cell-cell attachment, transmembrane domains, and important cytoplasmic domains cooperating with catenins (α, β and γ) for cytoskeletal attachment. This allows the formation of adherens junctions connecting epithelial cells and maintaining tissue stability [127]. Thus, E-cadherin is considered an important tumor suppressor implicated in several homeostatic signaling pathways and often down-regulated where the epithelial phenotype gives space to the mesenchymal phenotype of malignant cells [125]. E-cadherin favoring contact between neighboring cells was also found to play a role in inhibiting uncontrolled proliferation through the action of tyrosine kinase and receptor tyrosine kinase (RTK) [126,128]. Uncontrolled growth and resistance to programmed cell-death are associated with its loss in favor of N and P-cadherin [129]. Their upregulation and EMT suggest dissemination of cancer cells, greater invasive potential, and formation of metastasis. It was also found that cell-cell adhesion mediated by E-cadherin is an important suppressor of the Wnt/β-catenin pathway, while the switch to N-cadherin favors cell migration and resistance to programmed death through the activation of PI3K (Phosphoinositide-3-kinase) and MAPK/ERK (mitogen-activated protein kinase/extracellular signal-regulated kinases) pathways [126].

Interestingly, several studies observed how oncogenic viruses such as HPV-16, can induce E-cadherin downregulation in epithelial cells in favor of N-cadherin and EMT upregulation, favoring the formation of dysplastic lesions and cancer [129,130]. A study investigated the effect of various tyrosine kinase inhibitors (nilotinib, dasatinib, erlotinib, and gefitinib) on E-cadherin and β-catenin expression in both HPV-positive and negative HNSCC [131]. These small molecules do not act directly on these adhesion molecules but on EGFR and Wnt and only have a secondary effect on E-cadherin and β-catenin. Still, their use on HPV-positive cell lines caused a decrease in the previously dysregulated β-catenin expression, while both HPV-positive and negative cell lines showed patterns of E-cadherin increase.

### 3.2. Wnt/β-Catenin Pathway

The Wnt signaling pathway is a mechanism by which a range of glycoproteins called Wnt transduce signals from the outside to the inside of the cell through the action of β-catenin, a protein originally thought to be only involved in cell adhesion but now known to be also implicated in cell signaling and regulation of developmental and homeostatic processes [132,133]. This multifunctional protein is an important marker of EMT, and its expression goes hand-in-hand with that of E-cadherin; both adhesion molecules are abnormally expressed when the mesenchymal phenotype takes over. While the cadherin switch allows cancer invasiveness, the Wnt/b-catenin pathway plays a major role in cancer cell proliferation and differentiation [116,134]. While in normal epithelial cells, β-catenin closely cooperates with α-catenin, E-cadherin, and other adhesion molecules towards the stabilization of cell-cell adhesion, in cancer cells this pathway is atypically activated leading to abnormal expression and accumulation of β-catenin in the cytoplasm [135]. Its translocation to the nucleus interferes with transcription factors such as TCF/LEF (T-cell factor/lymphoid enhancing factor). The resulting complex activates expression of several genes involved in tumorigenesis, including c-myc, MMP-7 (matrix metalloproteinase-7), and VEGF [136]. Underlining the importance of the Wnt/b-catenin pathway is its association with APC (adenomatous polyposis coli), a tumor suppressor often mutated in cancer cells, found to be upregulated by Wnt1 and downregulated by β-catenin.

As previously mentioned, EMT is associated with several oncoviruses including HPV-16 and leads to the transformation of cells ultimately favoring cancer progression. A study by Rampias et al. reported that upon knock down of E6/E7 oncoproteins, there was a significant upregulation of Siah-1, a protein which normally promotes the degradation of β-catenin through the ubiquitin/proteasome system [137]. The effect was therefore a decrease in its nuclear levels and its effect on the previously mentioned oncogenes. Moreover, E7 binds to a component of APC, inhibiting its activity and therefore increasing β-catenin levels (Figure 2b) [61].

## 4. Immunology and Inflammation

As introduced in the previous section, it is well established that most tumors present an immunosuppressive microenvironment facilitating their growth and development [138]. This does not mean that no immune cells are present nor that those present are oblivious to the tumor’s presence. Instead, the inflammatory cells which infiltrate the tumor and detect tumor antigens to eliminate it (i.e., immune surveillance) end up helping its growth and spread [139]. These are usually immune suppressor cells, such as regulatory T lymphocytes (Treg) and myeloid-derived suppressor cells (MDSC), which normally regulate the immune system to avoid excessive harmful reactions (i.e., autoimmunity) [138]. Other cells can also be found, such as other T lymphocyte subsets, B lymphocytes, dendritic cells (DC), and macrophages [139].

Natural killer (NK) cells, which are a potent cytotoxic defense against pathogens and tumor cells and which respond to the downregulation of HLA antigens and MHC (major histocompatibility complex) class I induced by tumor cells to evade recognition, are rarely found in the TME [140]. This is another survival strategy enacted by the tumor, which purposely avoids NK cell recruitment [141]. In fact, in tumor infiltrating lymphocytes (TILs), studies showed an impaired recognition and response to antigens, as well as impaired cytokine secretion and recruitment of cytotoxic cells. Additionally, certain T cell subsets respond poorly to the persistent antigen stimulation caused by tumors and become exhausted over time, not responding effectively anymore. They are instead inhibited by downregulatory molecules such as PD-1, later discussed in the context of treatment strategies [139]. This has in many cases been associated with a worse prognosis but is still a source of great debate. CD4+ TCRαβ+ T helper cells (Th) are known to express the CD40L ligand that interacts with CD40 on dendritic cells, allowing secretion of several cytokines (IL-2, IL-15 and IFN γ) and activation of CD8+ cytotoxic T lymphocytes (CTLs), a T cell subset efficient in anti-tumor responses [142]. This, however, becomes exhausted and ineffective over time due to the down-regulatory effect of the TME, aided by TAMs (M2 type macrophages) and Tregs. This because CTLs only carry out their potent anti-tumor function after priming and activation by DCs, CD4+ T cells, and NK cells (which specifically respond to MHC class I downregulation, which “hides” tumors from other immune cells). CTLs then infiltrate the tumor to carry out their function. This step is not always allowed, so a treatment strategy might involve induction of priming and activation, possibly with immunotherapy. This allows to selectively target and sequester coinhibitory molecules on immune cells to allow binding of costimulatory molecules and activation [143]. This is known as immune checkpoint blockade (ICB), and an example could be targeting of PD-1/PD-L1 (programmed death receptor/programmed death ligand) or CTLA-4 with monoclonal antibodies such as nivolumab [138,144].

Saloura et al. investigated CTL infiltration in HPV-positive and negative HNSCC and analyzed the presence of immune checkpoints according to HPV status [142]. First, tumors were divided according to CTL infiltration and the presence of a 12-chemokine signature previously described as related to immune filtration and patient survival [145]. Tumors with high CTL infiltration were found to have a high chemokine signature, while less infiltrated tumors showed low chemokine levels, suggesting a relationship with patients’ prognosis. It was also found that the highly infiltrated tumors with a high chemokine signature were also HPV-positive, which is in line with the notion that HPV-positive tumors generally have a better prognosis. These also expressed the previously mentioned immune checkpoints, making them a possible target for the immunotherapy treatment approach.

### 4.1. NF-κB Pathway

Nuclear factor kappa B (NF-κB) is a family of transcription factors which bind the immunoglobulin (Ig) κ light chain enhancer element of B cells and hold a very important role in a range of physiological and immune functions. They were found to control cell growth and survival but also regulate immune responses and inflammatory processes [146]. Five members of the NF-κB family were identified, but the main one is a p65/p50 subunits heterodimer. These are usually inactive and bound to IκB (inhibitor of nuclear factor kappa B), released when NF-κB needs to be activated [147]. Recently, its role in tumorigenesis was established. NF-κB can be activated in two ways. The classical (canonical) pathway is dependent on IκB release through IKK and NEMO (NF-κB essential modulator) and is triggered by external stimuli (i.e., antigens, cytokines) picked up by TLRs (Toll-like receptors), ILRs (interleukin receptors), and TNFRs (Tumor-necrosis factor receptors). This could lead to the development of tumors or autoimmune disease. The alternative (non-canonical) pathway does not rely on the same components and is more likely to be tumor suppressive [148].

In cervical cancer, HPV E6/E7 oncoproteins downregulate NF-κB through a mutated IκB, preventing an immune response [149]. In this way, the virus can thrive, and the infection becomes chronic. After the formation of cancerous lesions, NF-κB is constitutively reactivated, probably by cytokines released by M2 macrophages in the TME [148,150,151]. The upstream mutations in signaling molecules, such as EGFR and RAS caused by the infection, result in dysregulated NF-κB function. This causes the expression of genes which lead to an aberrant growth and survival, such as cell immortalization and proliferation (i.e., telomerase genes, c-myc), as well as metastasis (i.e., EMT) and angiogenesis (i.e., VEGF) [147,152]. Its reactivation also induces expression of a family of proteins known to aid the development of cancer by causing genomic damage. These are called AID/APOBEC (activation induced cytodine deaminase) and target p53 and c-myc, which promote tumorigenesis when mutated [153]. Inhibition of this transcription factor might potentially present a solution to the problem of chemoradiotherapy resistance, no effective targeting strategy has however been uncovered for now.

### 4.2. JAK/STAT Pathway

The Janus Kinase/Signal Transducer and Activator of Transcription pathway (JAK/STAT), similarly to NF-κB, mediates signaling pathways controlling cell proliferation and survival (mainly STAT3 and 5) and also has a role in immune responses (STATs 1 and 2). JAK/STAT mediates signals (i.e., cytokines and growth factors) from transmembrane type I and II receptors directly into the nucleus, allowing a quick response to immune stimuli. Receptor dimerization induces phosphorylation of JAKs and of the receptor’s cytoplasmic tyrosine residue sequentially. STATs then bind and are phosphorylated to form dimers which translocate to the nucleus [154,155]. This mechanism also controls immune responses and in viral immunity. This happens mainly through the interferon signaling pathway involving STATs 1 and 2 and various receptors according to the type of IFN (I, II, III).

This pathway allows an anti-viral response which blocks its replication and spread, facilitating a response. HPV seems to interfere with this, and E6 and E7 may be implicated in a mutated pathway favoring the transcription of mutated genes and the development of cancer [156,157]. HPV in fact is capable of disrupting the STATs involved (1 and 2) in order to maintain its replication. This is achieved through binding of IRF9 (interferon 9 regulating factor) and inhibition of ISGF3 (interferon stimulated gene factor 3), normally translocated to the nucleus, preventing the expression of interferon stimulated genes (ISGs) essential to building an immune response. These STATs seem to play minor roles in tumorigenesis, and only STAT3 is truly considered an oncogene [158]. In the presence of E6, STAT3, after being activated mainly by the IL-6 cytokine family (it could also be activated by EGF and other cytokines and growth factors), mediates expression of genes such as VEGF and c-myc (also anti-apoptotic Cyclin-D and cell-cycle progression Bcl-xL), which play important roles in tumor growth, proliferation, and angiogenesis [154,159]. Even more relevant is the finding that a high HPV-16 viral load is associated with a high STAT3 phosphorylation (Figure 3). Similar results have been obtained for STAT5, suggesting that the inhibition of their associated pathways might represent a potential therapeutic target. IL-6 blockade using monoclonal antibodies might be promising to prevent activation of STAT3 in HPV-positive cells [158]. The use of small molecule inhibitors has been considered but has not, for now, yielded significant results due to their high toxicity; other strategies under testing involved the direct targeting of STAT3 mRNA with nucleotide therapeutics [160,161,162].

### 4.3. TGF-β/TNF-α

As seen, various cytokines are involved in the oncogenesis of several tumors, starting from the infection of cells by high-risk HPV (in the case of HPV-positive tumors) up to the dysregulation of many of the pathways involved [163]. Tumor necrosis factor alpha (TNF-α) is a 17kDa protein produced and secreted by several cells of the immune system, such as macrophages, NK cells, and T lymphocytes, to act as a pro-inflammatory cytokine. Its roles mainly relate to immune function during (i.e., leukocyte trafficking) and after infection (i.e., clearance of immune complexes). For this reason, it is clear that many things could go wrong in each part of this process, which ultimately lead to the formation of a tumor. TNF-α, together with other cytokines such as IL-10, represents part of the first line of defense against viruses, by its ability to induce a polarization of T cells to the specific subset needed. The Th1 subset of T helper lymphocytes (polarized by TNF-α) is the preferred choice in this case as its proinflammatory function can effectively prevent viral replication and clear the infection; defective polarization instead, leads to the immunosuppressive Th2 subset (polarized by IL-10), which could lead to tumor formation. It is also worth noting, however, that chronic inflammation is also associated with the formation of precancerous and cancerous lesions [163].

The exact role of TNF-α in tumor development is not entirely clear. Some studies do indeed suggest a beneficial role of the Th1 polarization in HPV infections, suggesting that polarization prevents replication and the transformation of infected cells through apoptosis. Other studies involving HPV-16 and 18, however, suggest a viral resistance and thus an irrelevant effect of TNF-α [164,165]. Some studies even suggested that TNF-α might play a role in the influence of E7 on NF-κB through the inhibition of IκB [164]. Similar results were obtained in a study investigating the effect on NF-κB of TNF-α in the presence of TGF-β [166]. This combination allegedly induced EMT, migration, and self-renewal in HeLa cells through the specific action of the NF-κB/Twist axis. Another study investigated the presence in OPSCC (oropharyngeal squamous cell carcinoma) patients of various pro and anti-inflammatory cytokines known to play a role in HPV-infection and tumorigenesis [167]. The cytokines included were TNF-α, TGF-β, and IL-10 as well as IFN-γ and VEGF, and they were tested through the collection of patients’ saliva samples. Their elevated presence was confirmed in patients, especially in patients where HPV was also detected, as opposed to controls. These results suggest possible future diagnostic applications, treatment strategies, and prognosis indicators.

## 5. miRNAs 

Microribonucleic acids (miRNAs) are small (19–25 nucleotides long) single stranded RNAs, also found in cervical cancer and HNSCC, that are both HPV-positive and HPV-negative (reviewed in [168,169,170]). miRNA are non-coding and can only interfere with RNA by altering its expression towards an oncogenic or tumor suppressor function [170]. Their expression also changes throughout the various phases of tumorigenesis [171]. As illustrated in Table 2, miRNAs can be both markers for diagnosing HPV-positive tumors, as prognostic indicators for response to therapy, as well as new targets for treatment strategies (described in [172]). Targeting miR-21 and 7a expression, for example, could positively influence STAT3 and lead to a correct functioning of the JAK/STAT pathway [173]. Sannigrahi et al. found that upon upregulation of the HPV-inhibited Hsa-miR-139-3p, p53 function can be restored and chemoresistance can be reverted [174].

## 6. DNA Damage Response

The DNA damage response (DDR) is a repair mechanism found in cells which detects damaged DNA, adjusting it and preventing its duplication [180]. The major repair mechanism ensuring genomic integrity in human cells is formed by a group of kinases belonging to the PI3-K-related kinase family (PIKK): ATR, ATM, and DNA-PK (DNA-dependent protein kinase). Each can contribute to restore the correct DNA sequence through different mechanisms. DNA-PK, for instance, relies on non-homologous end-joining (NHEJ) to ligate double-stranded breaks (DSB) without using a homologous template. Thus, it is more error prone than ATM, which relies on homologous recombination. ATK is instead needed to repair single-stranded breaks [181]. These kinases generally act by phosphorylating several downstream effector proteins like, for example, BRCA1 and CHK1 in the case of ATM, and CHK2 in the case of ATR [180].

Oncogenic viruses have been found to interfere with and modulate various components of these DDR pathways in order to survive and replicate in host cells. HPV uses E2 to identify an origin of replication and recruit the E1 helicase. E6 and E7 then allow the virus to degrade important regulatory proteins in order to interfere with the cell-cycle; re-entry into the S-phase allows amplification of the viral DNA [182]. Several changes were found to be induced in HPV-positive cells, such as the HPV-mediated activation of the ATM and ATR pathways in order to favor viral replication [180]. Specifically, important damage regulatory proteins such as BRCA1, FANCD2, and γH2AX are activated. Recently, it was shown that HPV infection induces DNA damage and correlates with cervical precancerous lesions and cancer [183]. Transformation and immortalization of keratinocytes is thought to be likely induced by the high level of genomic instability caused by E6 and E7 [184]. These changes could potentially represent a starting point for developing new targeted therapies. A study found a relationship between HPV status and radiosensitivity [185]. HPV-positive OPSCC cells in fact showed increased levels of several proteins involved in single stranded DNA repair (as well as base excision repair), such as PARP-1, DNA polymerase β, PNKP, and XRCC1. Treatment with Cidofovir, an antiviral agent targeting viral DNA polymerases, showed a decreased cellular growth due to increased levels of DNA repair proteins and γH2AX, as well as an arrest in the cell-cycle at S and G2/M [186].

## 7. DNA Methylation

DNA methylation is an epigenetic change consisting in the covalent modification of a strand of DNA through the addition of methyl groups. This is accomplished and maintained through the action of several enzymes called DNA methyltransferases. In humans, this involves the 5′ position of cytosine rings placed at 5′ from guanine bases, thus forming CpG islands (reviewed in [187]). Like other epigenetic changes, methylation results in altered gene expression without changing the DNA sequence [188].

DNA hypermethylation was reported in both host and HPV genes of tumor cells and was recently proposed as a biomarker for cervical cancer as well as for other HPV-related cancers such as HNSCC [188,189]. The most widely studied of these alterations interests the CpG islands located in the promoter region of genes, such as tumor suppressor genes. The DNA methylation of 5-cytosine at CpG dinucleotides leads to the silencing of these tumor suppressor genes and activation of oncogenes, with a resulting carcinogenic effect [189]. Moreover, the presence of this modification in the HPV upstream regulatory region (URR) E2-binding sites (E2BS) seems to play a pivotal role in the carcinogenic transformation of squamous cells [190]. It is not entirely clear what the initial trigger for hypermethylation is, although we know that the low levels of E2 resulting from it allow E6 and E7 overexpression, which in turn promote DNA methyltransferase 1 (DNMTI), leading to more methylation and cancer development. It is worth noting that hypermethylation has also been found in normal cells close to HPV-related lesions [191,192].

In a screening of various populations of HPV-positive women, in which all HR-HPVs were included, CpG islands of L1 and L2 genes were analyzed in several cervical cancer types. The results showed a strong correlation between positivity to HPV (independent of HPV type), methylation, and cancer risk, suggesting the possible future application of methylation assays in diagnostic cancer screens for HPV-positive patients [193]. Similarly, gargle and biopsy samples from OPSCC male patients were analyzed for HPV-status and type as well as methylation at various CpG sites in order to understand the potential for early cancer detection through analysis of methylation biomarkers. A strong correlation was observed between OPSCC biopsy samples and gargles, suggesting a potential early, non-invasive screening method [194].

## 8. Conclusions

HPV tumorigenesis is quite complex. Understanding all the events which take place during this process is important to help future research towards the development of effective therapies against HPV-tumors. Here we showed a variety of mechanisms by which HPV can impinge cellular pathways for its own needs. It can interact with and change the activity of many proteins.

Processes altered by HPV ranged from its most known targets, p53 and pRb inhibition dependent pathways, to signaling pathways such as EGFR and MAPKs. HPV can also tailor energy and metabolism by targeting Akt, mTOR, and autophagy. DNA damage response is pushed by HPV to obtain an efficient viral replication. Epithelial-mesenchymal transition is achieved by HPV, as well as deregulation of immune and inflammatory responses. We also reviewed many miRNAs that are regulated by HPV and interfere with a plethora of cellular pathways. HPV also upregulates DNA methylation, leading to inhibition of tumor suppressors.

Among these cellular processes, we highlighted several potential targets for HPV therapies, but further studies are needed to understand which can evolve from a potential target to a real targeted therapy for HPV-tumors. We have also discussed potential biomarkers involved in pathways that are differentially affected or modified during the onset of HPV-related cancers, such as Akt, mTOR, miRNAs, TNF-α, TGF-β, BRCA1, and FANCD2. However, the translation to the clinics of potential prognostic and/or diagnostics biomarkers remains a challenge. In this respect, in order to advance our understanding, it will be crucial in the next years to develop strong preclinical models and increase the number of clinical studies.

Importantly, HPV vaccination is the most important prevention effort towards the eradication of HPV-driven diseases. These campaigns are already ongoing worldwide (reviewed in [195]).

## Figures and Tables

**Figure 1 pathogens-10-00262-f001:**
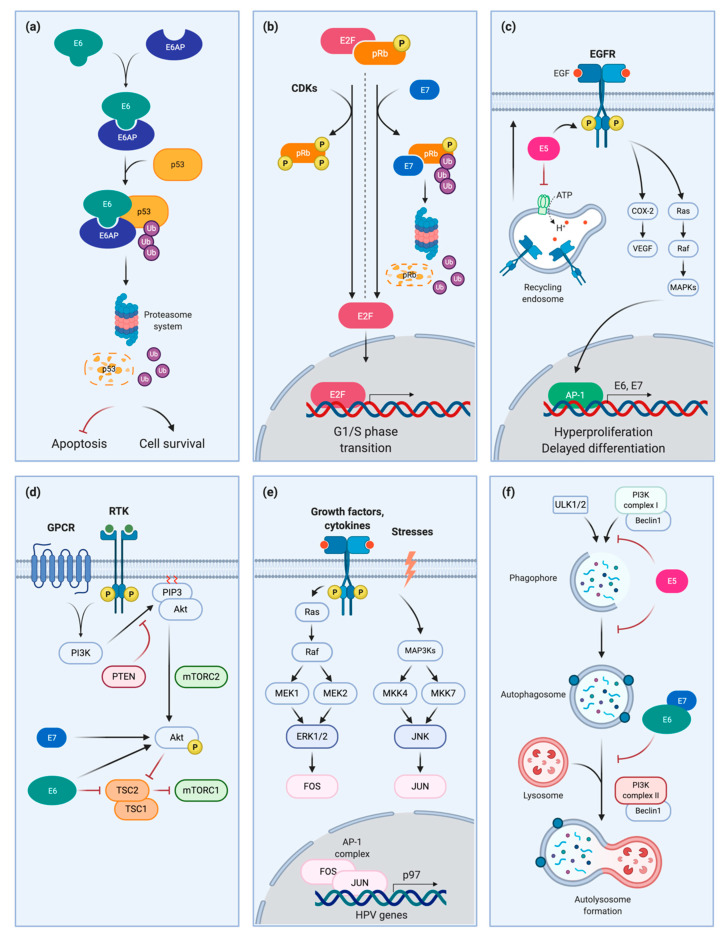
Signaling pathways impinged by HPV: (**a**) HPV E6-mediated degradation of p53; (**b**) pRb degradation and inactivation by HPV E7; (**c**) EGFR regulation by HPV E5; (**d**) PI3K, Akt and mTOR are deregulated by HPV; (**e**) HPV gene transcription induced by AP-1; (**f**) Autophagy inhibition by HPV oncoproteins.

**Figure 2 pathogens-10-00262-f002:**
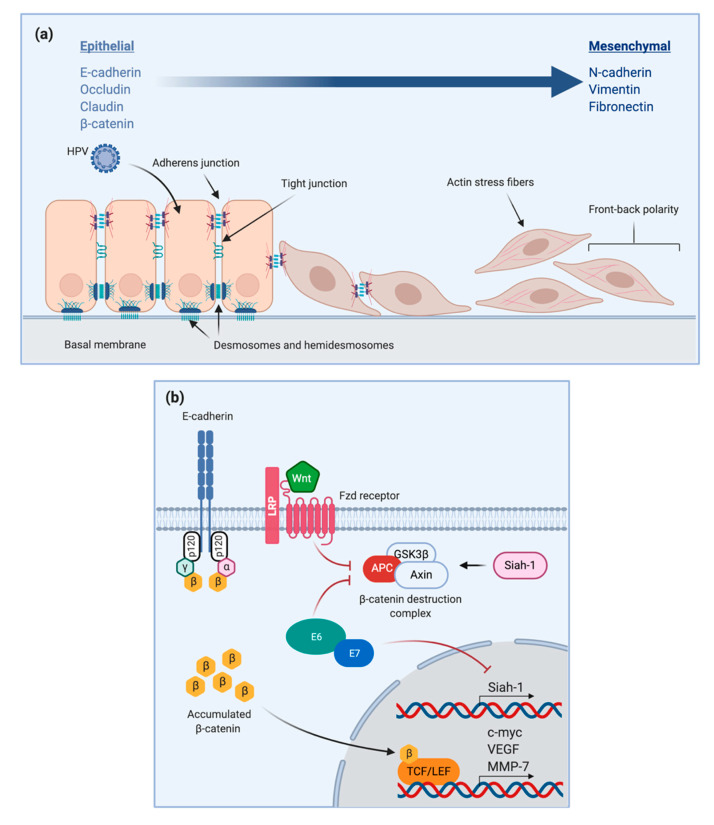
EMT and HPV (**a**) Changes in epithelial and mesenchymal markers induced by HPV; (**b**) HPV16 E6/E7 inhibit Siah-1 expression and APC activity, leading to β-catenin upregulation.

**Figure 3 pathogens-10-00262-f003:**
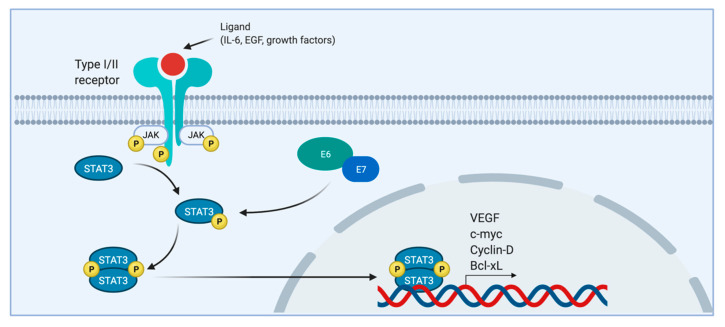
HPV 16 E6/E7 induce the phosphorylation of STAT-3 and the expression of proliferation, angiogenesis, and tumor growth genes.

**Table 1 pathogens-10-00262-t001:** Main HPV types and their associated diseases.

	Low Risk	High Risk
**HPV Type**	1, 6, 10, 11, 32, 42, 44	16, 18, 31, 33, 35, 39, 45, 51, 52, 56, 58, 59, 66, 68
**Associated Disease**	Anogenital warts Cutaneous warts Recurrent respiratory papillomatosis Heck’s disease	Intraepithelial neoplasia Invasive carcinoma: HNSCC, Cervical cancer, Anogenital cancers, Non-melanoma skin cancer

**Table 2 pathogens-10-00262-t002:** Main features of miRNA commonly dysregulated in HPV-positive cancers.

miRNA	Viral Oncoprotein	Pathways Involved	Role in Tumorigenesis	References
**miR-200a**	HPV-16 E6/E7	EMT	Downregulated	Wang et al., 2019 [175]
			Downregulation prevents EMT inhibition	Eades et al., 2011 [176]
**miR-9**	HPV-16 E6	Cell metabolism	Upregulated in recurring HNSCC and cervical cancer	Božinović et al., 2019 [177]
**miR-7a, miR-21**	HPV-16 E6	JAK/STAT	Maintain STAT3 activated in HPV-positive cells	Shishodia et al., 2015 [173]
**miR-29**	HPV-16 E6/E7		Often downregulated.	Božinović et al., 2019 [177]
**miR-218**	HPV-16 E6	PI3K/Akt pathway, cell-cycle	Upregulates expression of the LAMB3 epithelial cell marker.	Zheng et al., 2013 [178]
**miR-34a**	HPV-16 E6	p53-dependent pathway	Downregulated	Zhang et al., 2016 [179]
**Hsa-miR-139-3p**	HPV-16 E1, E6/E7	p-53, cell-cycle	Upregulation restores p53 expression and inhibits E6/E7.	Sannigrahi et al., 2017 [174]

## Data Availability

Data sharing not applicable.

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
