# Peer review of "Human Papillomavirus and Cellular Pathways: Hits and Targets"

_pathogens, 2021, doi:10.3390/pathogens10030262_

Round 1

Reviewer 1 Report

Recommendation: Major revisions

Comments: 

This manuscript describes the molecular mechanisms and pathways affected by HPV and their potential features for new biomarkers or targeted therapy. The authors need to address the following comments and revise the manuscript accordingly. 

  1. Please consider to include a separate paragraph on the initiation and maintenance of HPV-associated cancers.
  2. Page2, line 58: E6 is able to increase telomerase activity by up- regulation of telomerase reverse transcriptase (TERT); please highlight “Telomerase activation”.
  3. Consider to provide a table for HPVs and human diseases.
  4. Methylation of HPV DNA has been proposed as a novel biomarker. Consider to highlight and include the following references. 
    • Clarke MA, Gradissimo A, Schiffman M, Lam J, Sollecito CC, Fetterman B, Lorey T, Poitras N, Raine-Bennett TR, Castle PE, Wentzensen N, Burk RD. Human Papillomavirus DNA Methylation as a Biomarker for Cervical Precancer: Consistency across 12 Genotypes and Potential Impact on Management of HPV-Positive Women. Clin Cancer Res. 2018 May 1;24(9):2194-2202. doi: 10.1158/1078-0432.CCR-17-3251. Epub 2018 Feb 2. PMID: 29420222; PMCID: PMC5932258.
    • Roy, D.; Tiirikainen, M. Diagnostic Power of DNA Methylation Classifiers for Early Detection of Cancer. Trends Cancer 2020, 6, 78–81. DOI:https://doi.org/10.1016/j.trecan.2019.12.006.
  1. Consider to cover and highlight the biomarkers for prediction, detection and improved prevention efforts.

Author Response

We thank the Reviewers for their suggestions and comments. We believe we have addressed all their criticisms, as detailed in our replies in bold.

REVIEWER 1

This manuscript describes the molecular mechanisms and pathways affected by HPV and their potential features for new biomarkers or targeted therapy. The authors need to address the following comments and revise the manuscript accordingly. 

  1. Please consider to include a separate paragraph on the initiation and maintenance of HPV-associated cancers.

Thank you for this suggestion, we included this paragraph in the introduction.

  1. Page2, line 58: E6 is able to increase telomerase activity by up- regulation of telomerase reverse transcriptase (TERT); please highlight “Telomerase activation”.

We included a sentence about TERT in the Introduction: please see lines 58-59.

  1. Consider to provide a table for HPVs and human diseases.

Thank you for this suggestion, we included a novel Table, Table 1, describing the main HPV types divided in LR and HR, and the related diseases.

  1. Methylation of HPV DNA has been proposed as a novel biomarker. Consider to highlight and include the following references. 
  • Clarke MA, Gradissimo A, Schiffman M, Lam J, Sollecito CC, Fetterman B, Lorey T, Poitras N, Raine-Bennett TR, Castle PE, Wentzensen N, Burk RD. Human Papillomavirus DNA Methylation as a Biomarker for Cervical Precancer: Consistency across 12 Genotypes and Potential Impact on Management of HPV-Positive Women. Clin Cancer Res. 2018 May 1;24(9):2194-2202. doi: 10.1158/1078-0432.CCR-17-3251. Epub 2018 Feb 2. PMID: 29420222; PMCID: PMC5932258.
  • Roy, D.; Tiirikainen, M. Diagnostic Power of DNA Methylation Classifiers for Early Detection of Cancer. Trends Cancer2020, 6, 78–81. DOI:https://doi.org/10.1016/j.trecan.2019.12.006.

We added this new section, section 7 and referenced these manuscripts accordingly.

  1. Consider to cover and highlight the biomarkers for prediction, detection and improved prevention efforts.

We added a sentence in the Introduction on how to detect HPV in cancers, precisely in lines 74-79.

Furthermore, we expanded the Conclusions section to provide a wrap-up sentence on the discussed biomarkers for prognosis/diagnosis. Finally, in this same section we also mentioned vaccination as the most important prevention effort towards the eradication of HPV-driven diseases.

Reviewer 2 Report

A well written review. It will help to get a fast overview in this field of research. The main mechanisms discussed in carcinogenesis of HPV are discussed and references for further studies are given. Graphics are helping in the understanding of these mechanisms.

There are no major comments from my side.    

There are a few minor points only from my side:
1. Are the figures all made by Alice Viotti? If yes: Very nice work. If not: Do not forget to give the source of the figure. I recommend to be accurate in this point (I assume you are, nevertheless check it).
2. Table 1 seems to be a little long. Could you make a change here, if possible?
3. Spelling and grammar: Is the paper edited by a professional service (since I am not a native speaker, I am not able to judge about it)?

Author Response

REVIEWER 2

A well written review. It will help to get a fast overview in this field of research. The main mechanisms discussed in carcinogenesis of HPV are discussed and references for further studies are given. Graphics are helping in the understanding of these mechanisms.

There are no major comments from my side.    

There are a few minor points only from my side:
1. Are the figures all made by Alice Viotti? If yes: Very nice work. If not: Do not forget to give the source of the figure. I recommend to be accurate in this point (I assume you are, nevertheless check it).

Alice Viotti, Daria Duca and Alessandro Medda made the figures using biorender.com. We acknowledged this, as asked by the developers.

  1. Table 1 seems to be a little long. Could you make a change here, if possible?

We reduced descriptions of miRNA in the table. Thank you for your suggestion.

  1. Spelling and grammar: Is the paper edited by a professional service (since I am not a native speaker, I am not able to judge about it)? 

Thank you, yes we checked for grammar and spelling.

Reviewer 3 Report

This manuscript is generally a well-written and comprehensive overview of cellular pathways and processes impacted by high-risk HPV infection. This report will be a useful starting point for those investigators seeking specific HPV oncoprotein-cell interactions that could be exploited for clinical purposes as it gives a brief, cogent summary of the known interactions.

This is a comprehensive and thorough review of cellular pathways and processes related to HPV carcinogenesis, but is less well developed about the virus itself. While this manuscript is already fairly lengthy, expanded coverage of viral biology and the properties of the viral oncoproteins in the introduction would be useful for the non-HPV specialist. Similarly, in several subsections the pathways were clearly and extensively described while effects of the viral oncoproteins were simply noted without conclusions or speculations about their actual contribution in the oncogenic process. More emphasis in the sections should be given to mechanistic rather than observational references to the effect of oncoproteins on the pathways in question.

Author Response

REVIEWER 3

This manuscript is generally a well-written and comprehensive overview of cellular pathways and processes impacted by high-risk HPV infection. This report will be a useful starting point for those investigators seeking specific HPV oncoprotein-cell interactions that could be exploited for clinical purposes as it gives a brief, cogent summary of the known interactions.

This is a comprehensive and thorough review of cellular pathways and processes related to HPV carcinogenesis, but is less well developed about the virus itself. While this manuscript is already fairly lengthy, expanded coverage of viral biology and the properties of the viral oncoproteins in the introduction would be useful for the non-HPV specialist.

We expanded viral infection and HPV proteins description in the Introduction. We hope you now find this section helpful for non-HPV-specialists.

 Similarly, in several subsections the pathways were clearly and extensively described while effects of the viral oncoproteins were simply noted without conclusions or speculations about their actual contribution in the oncogenic process. More emphasis in the sections should be given to mechanistic rather than observational references to the effect of oncoproteins on the pathways in question.

We tried to give more emphasis to the mechanistic part and relative contribution to carcinogenesis when possible. Indeed, in some cases such as HPV-mediated E-cadherin regulation, the molecular mechanisms are still not well understood. However, if you would like us to provide more details in some parts of our review, please specify where.

Round 2

Reviewer 1 Report

Comments: 

The authors have addressed all the comments quite thoroughly and this version of the manuscript is improved. Please publish.